

# Stratification and overturning circulation are intertwined controls on ocean heat uptake efficiency in climate models

Linus Vogt[1], Jean-Baptiste Sallée[1], and Casimir de Lavergne[1]

[1]Sorbonne Université, CNRS/IRD/MNHN, Laboratoire d'Océanographie et du Climat Expérimentations et Approches Numériques (LOCEAN), Paris, France

**Correspondence:** Linus Vogt (linus.vogt@locean.ipsl.fr)

**Abstract.** The global ocean takes up over 90% of the excess heat added to the climate system due to anthropogenic emissions, thereby buffering climate change at the Earth's surface. A key metric to quantify the role of the oceanic processes removing this heat from the atmosphere and storing it in the ocean is the ocean heat uptake efficiency (OHUE), defined as the amount of ocean heat uptake per degree of global surface warming. Despite the importance of OHUE, there remain substantial uncertainties

concerning the physical mechanisms controlling its magnitude in global climate model simulations: ocean mixed layer depth, Atlantic meridional overturning circulation (AMOC) strength, and upper ocean stratification strength have all been previously proposed as controlling factors.

In this study, we analyze model output from an ensemble of 28 climate models from the Coupled Model Intercomparison Project, phase 6 (CMIP6), in order to resolve these apparently divergent explanations. We find that stratification in the mid-

latitude Southern Ocean is a key model property setting the value of OHUE due to its influence on Southern Ocean overturning. The previously proposed role of the AMOC for OHUE is explained by a linkage of stratification model biases between the subpolar North Atlantic and the Southern Ocean. Our analysis thus reconciles previous attempts at explaining controls on OHUE, and highlights the importance of interlinked model biases across variables and geographical regions.

## 1   Introduction

The global ocean buffers anthropogenic climate change by taking up excess heat and carbon from the atmosphere. Since the preindustrial era, over 90% of the additional heat that has entered the Earth system as a result of changes in the Earth's radiative balance has been stored in the ocean (von Schuckmann et al., 2020; Forster et al., 2021). This ocean heat uptake (OHU) is a key process determining the sensitivity of the climate system to external perturbations, in particular to radiative forcing from increased atmospheric greenhouse gas concentrations.

More than half of the observed increase in ocean heat content (OHC) is concentrated in waters shallower than $700\,\mathrm{m}$ depth (von Schuckmann et al., 2020). Under increased radiative forcing, anomalous air-sea heat fluxes enter the ocean through its surface and quickly warm the ocean mixed layer on seasonal to interannual timescales, whereas the deep ocean (below $2000\,\mathrm{m}$ depth) is more isolated from the atmosphere and is warmed on timescales of decades to centuries (Cheng et al., 2022). Heat is fluxed towards the deep ocean through a multitude of processes, including subduction from the mixed layer (Marzocchi et al.,



2021), mean downwelling flows and vertical mixing (Exarchou et al., 2015), and (sub-)mesoscale eddy processes contributing notably to isopycnal mixing (Gregory, 2000; Morrison et al., 2016).

A key metric to quantify the efficiency of these processes at hiding heat from the atmosphere under transient climate change is the *OHU efficiency* (OHUE), defined as the rate of global OHU per degree of global mean surface warming (e.g., Gregory and Mitchell, 1997; Gregory et al., 2023), with units of $\mathrm{Wm^{-2}K^{-1}}$:

$$\mathrm{OHUE} = \mathrm{OHU}/\Delta T, \tag{1}$$

where OHU is the increase in OHC relative to preindustrial levels expressed as a flux of energy per unit global surface area, and $\Delta T$ is the global mean surface air temperature anomaly relative to preindustrial levels.

In global climate model (GCM) simulations of transient climate change, OHUE estimates span a factor of two across different models (Gregory et al., 2023), due to inter-model spread in both OHU (e.g., Vogt et al., 2024) and transient surface
warming projections (e.g., Meehl et al., 2020). In an attempt to determine the source of this uncertainty and to find potential observational constraints on OHUE, previous studies have proposed a number of oceanic metrics that control OHUE in GCMs participating in successive phases of the Coupled Model Intercomparison Project (CMIP; Eyring et al., 2016). High-latitude ocean mixed layer depths were first identified as a possible control of transient warming rates in the ocean and atmosphere using the CMIP3 ensemble (Boé et al., 2009). Subsequently, the strength of the Atlantic Meridional Overturning Circulation (AMOC)
in the preindustrial baseline climate has been found to correlate well with OHUE across CMIP5 multi-model ensembles (Kostov et al., 2014; Winton et al., 2014) as well as across parameter perturbation ensembles (Romanou et al., 2017; Saenko et al., 2018) and initial condition ensembles (He et al., 2017), each based on a single model. However, the actual amount of anomalous heat entering the North Atlantic and being subducted by the AMOC is small compared to the OHU occurring in the mid-latitude Southern Ocean (Frölicher et al., 2015; Cheng et al., 2022). This is explained by aerosol-induced cooling in the North
Atlantic and higher subduction rates in the Southern Ocean (Williams et al., 2024). Furthermore, OHUE actually decreases when the AMOC strengthens under transient forcing (Stolpe et al., 2018). Gregory et al. (2023) have thus postulated that the correlation between AMOC and OHUE may originate from a common dependence on a third factor that would characterize the preindustrial ocean state of a model and influence both AMOC and OHUE.

A promising candidate that potentially controls both AMOC and OHUE is the strength of the upper ocean stratification
(Kuhlbrodt and Gregory, 2012), i.e. the density difference between the upper and deeper ocean, which is the main reason for the deep ocean's relative isolation from other parts of the climate system. Because large-scale ocean currents and smaller-scale mixing processes occur preferentially along isopycnal surfaces, stratification impedes the exchange of properties between the upper and deep oceans (e.g., McDougall et al., 2014). Recent studies have highlighted the impact of upper ocean stratification on OHUE in GCMs. Bourgeois et al. (2022) constrained oceanic heat and carbon uptake in the Southern Ocean using observed
and CMIP6-simulated stratification profiles in the region between $30°$S and $55°$S. Similarly, Liu et al. (2023) underscored the importance of salinity stratification in influencing OHUE in CMIP6 models and used global sea surface salinity observations to estimate OHU efficiency through an emergent constraint. Finally, Newsom et al. (2023) showed that the depth of the global





pycnocline, used as a metric to quantify upper ocean stratification, is strongly correlated with OHUE across CMIP5/6 models and across a parameter perturbation ensemble of a single model.

It remains unclear, however, how to reconcile these proposed OHUE controls based on AMOC strength, mixed layer depth (MLD), and stratification. This is not least due to the fact that these variables are interconnected: a deeper mixed layer translates to reduced stratification and vice versa, and North Atlantic MLD and stratification condition the AMOC (Jackson et al., 2023; Nayak et al., 2024). Furthermore, climate model biases can be linked between remote regions of the Earth (Wang et al., 2014; Luo et al., 2023), complicating the analysis and interpretation of regional climate metrics in GCMs. For instance, the

extratropical oceans, in particular the subpolar North Atlantic and the Southern Ocean, have an outsize role in ventilating the global ocean and in storing heat and carbon (Frölicher et al., 2015; Shi et al., 2018). In these regions, the stratification is directly related to the large-scale global ocean circulation since the upper and deep oceans are connected via upwards-sloping isopycnals (Kuhlbrodt et al., 2007; Kamenkovich and Radko, 2011; Morrison et al., 2022). A potential link between Southern Ocean and subpolar North Atlantic stratification could therefore provide insight into the control of upper ocean stratification

on OHUE in GCMs.

  In this study, we use an ensemble of CMIP6 models under idealized $CO_2$ forcing as well as a global ocean state estimate in order to analyze the inter-model relationships and biases in upper ocean properties (stratification and mixed layer depth) and meridional overturning metrics (AMOC and Southern Ocean overturning strength), as well as their combined influence on OHUE.

In particular, we aim to answer the following questions:

   – In which oceanic regions does stratification control OHUE?

   – How do biases in temperature and salinity stratification differ in their control on OHUE?

   – What explains the positive correlation between AMOC strength and OHUE across CMIP6 models?

   – What is the role of meridional overturning in the Southern Ocean for OHUE?

The remainder of this article is organized as follows: In Sect. 2, we present the data and methods used in this study. In Sect. 3, we analyze the dependence of OHUE on upper ocean properties and meridional overturning metrics both from a global (Sect. 3.1) and a local perspective (Sect. 3.2). In Sect. 4, we then present the inter-model relationships between these upper ocean properties on one hand and the meridional overturning metrics on the other hand. In Sect. 5, we analyze the ensemble mean and inter-model spread of historical stratification and its bias relative to observations, including a link between GCM

stratification biases between the Southern Ocean and the subpolar North Atlantic (Sect. 5.2). Finally, in Sect. 6 we offer a schematic picture of all major inter-model relationships explored in this study and conclude by answering the five questions posed above.



## 2 Methods

### 2.1 CMIP6 model output

We use model output from a set of 28 climate models from 14 modeling centers run in two CMIP6 experiments: a baseline experiment with preindustrial forcings (piControl experiment), and a perturbed scenario forced by an idealized $CO_2$ increase of 1% per year during 150 years (1pctCO2 experiment). We use one ensemble member per model, with the 1pctCO2 run branching off from the piControl run (Table A1). All model output used for the analysis (principally ocean potential temperature and ocean salinity) is regridded onto a regular $1° \times 1°$ latitude–longitude grid in order to allow the calculation of local inter-model correlations at each grid cell. Anomalies of variables in the 1pctCO2 experiment relative to the piControl run are calculated by subtracting the appropriate piControl period from the 1pctCO2 data; since piControl runs are extended over the 150-year period of the 1pctCO2 experiment, this method removes the effect of model drift.

### 2.2 Calculation of ocean variables

Ocean heat content per unit volume is defined as $OHC = \rho_0 C_p \theta$, where $\rho_0 = 1035$ kg m$^{-3}$ is a reference density, $C_p = 3992$ J kg$^{-1}$ K$^{-1}$ is a reference heat capacity (as defined in TEOS-10, e.g., Griffies et al., 2016), and $\theta$ is potential temperature. Global OHU in the 1pctCO2 experiment is then calculated as the time derivative of the three-dimensional integral of the OHC anomaly relative to the preindustrial state.

Ocean heat uptake efficiency (OHUE) is defined as in Gregory et al. (2023): the total OHU divided by 1.5 times the global mean sea surface temperature anomaly at years 60–80 in the 1pctCO2 run, which is the 20-year period around the time of $CO_2$ doubling relative to the preindustrial.

The AMOC strength is calculated using the overturning streamfunction variables in latitude–depth coordinates from the CMIP6 output and is defined as the streamfunction maximum in the Atlantic basin at 26.5°N and below 500 m depth.

Stratification is defined as the squared buoyancy frequency $N^2$ integrated in depth between 0 and 1500 m, resulting in units of m s$^{-2}$. The squared buoyancy frequency $N^2$ is calculated using the TEOS-10 software toolbox (McDougall and Barker, 2011). The depth of 1500 m is chosen to encompass the mixed layer as well as the internal pycnocline (Gnanadesikan, 1999; Klocker et al., 2023). The main results of this study are tested with different values of this maximal depth (spanning a range from 400 m to 2500 m) and will be shown to be only weakly sensitive to this particular choice. The stratification is further decomposed into contributions from temperature and salinity, according to

$$N^2 = N_T^2 + N_S^2 = -\alpha \frac{\partial \theta}{\partial z} + \beta \frac{\partial S}{\partial z} \tag{2}$$

where $\alpha$ is the thermal expansion coefficient, $\beta$ the haline contraction coefficient, and $S$ salinity. The sum of these two terms reproduces the total $N^2$ exactly.

Mixed layer depth is defined as the minimum depth where the monthly potential density $\sigma_0$ deviates by 0.03 kg m$^{-3}$ from its value at 5 m depth (de Boyer Montégut et al., 2004). For consistency, this definition is used even for models that have the MLD variable `mlotst` available as part of their CMIP output.





To calculate the strength of the upper Southern Ocean overturning cell, we first calculate the time-mean overturning stream-function in latitude–density coordinates from time-mean meridional ocean velocity and potential density referenced to 2000 dbars ($\sigma_2$) (e.g. Farneti et al., 2015):

$$\overline{\Psi}_{\mathrm{SO}}(y, \sigma_2) = -\int \mathrm{d}x \int_{-H}^{\bar{z}(x,y,\sigma_2)} \bar{v}(x, y, z') \, \mathrm{d}z' \mathrm{d}x, \qquad (3)$$

where $x$, $y$, and $z$ are longitude, latitude, and depth; $H(x, y)$ is the depth of the ocean bottom; $v$ is residual mean meridional

mass transport (CMIP variable `vmo`, including resolved and parameterized transport); and $\bar{z}(x, y, \sigma_2)$ is the local depth of the isopycnal $\sigma_2$. The strength of the upper cell $M_{\mathrm{SO}}$ is then defined as the time-mean streamfunction maximum within the $1034\ \mathrm{kg}\ \mathrm{m}^{-3} < \sigma_2 < 1038\ \mathrm{kg}\ \mathrm{m}^{-3}$ density range and between 35°S and 40°S.

    For a complementary quantification of Southern Ocean overturning, we compute surface flux water mass transformation (SFWMT), a measure of overturning inferred from surface buoyancy fluxes, following e.g. Jackson and Petit (2023). The

SFWMT is the derivative of the surface buoyancy flux into the Southern Ocean south of 30°S with respect to density:

$$\Psi(\sigma_2) = \frac{\partial}{\partial \sigma_2} B(\sigma_2), \qquad (4)$$

where the surface buoyancy flux is a sum of heat and freshwater terms:

$$B(\sigma_2) = -\alpha \frac{Q}{C_p} - \beta \frac{\sigma_2 s W}{1 - s}. \qquad (5)$$

In this equation, $s$ is non-dimensional sea surface salinity, and $W$ is the surface freshwater flux (CMIP variable `wfo`) in units

of $\mathrm{kg}\ \mathrm{m}^{-2}\ \mathrm{s}^{-1}$. As a single measure of Southern Ocean overturning strength inferred from surface buoyancy fluxes, we choose the difference

$$M_{\mathrm{WMT}} = \max_{\sigma_2} \Psi - \min_{\sigma_2} \Psi. \qquad (6)$$

## 2.3   Observation-based data

For comparison of model fields with observationally constrained data, we use potential temperature and salinity data from the

ECCO Version 4 global state estimate (ECCO Consortium et al., 2024; Forget et al., 2015) with data coverage from 1992 to 2017. To calculate stratification strength and MLD, the ECCO output fields are regridded and processed in the same way as the CMIP6 model output.

## 2.4   Inter-model empirical orthogonal function analysis

An empirical orthogonal function (EOF) algorithm (Dawson, 2016) is applied to two-dimensional model fields to construct

inter-model EOF patterns, expressed as the correlation across models between the principal component value and the input field at each grid cell. This corresponds to a standard EOF analysis, but with the variance maximized by each EOF being measured across models instead of in time (e.g. Hu et al., 2020).





For the EOF analysis of preindustrial mixed layer depth (Fig. A8), a number of outlier models with extreme values of the first principal component were identified and removed from the analysis in order to facilitate interpretation. For this, the EOF

algorithm was iteratively applied five times to the preindustrial annual mean MLD fields of all models and the model with the most extreme value of the first principal component was removed.

### 2.5 Classification of vertical stratification profiles

An unsupervised ocean profile classification algorithm (Maze et al., 2017; Maze, 2020) is applied to vertical profiles of $N_T^2$ and $N_S^2$ to obtain a pre-specified number of 8 representative classes characterized by the shape and amplitude of temperature

and salinity stratification profiles. As input to the classification procedure, the preindustrial time-mean $N_T^2$ and $N_S^2$ profiles are pooled together from all grid cells and from all models.

## 3 Global and local controls on ocean heat uptake efficiency

We begin by investigating the main proposed controls on OHUE in our set of 28 CMIP6 GCMs in the preindustrial state. These variables belong to two categories: upper ocean properties (i.e., stratification and mixed layer depth), and meridional

overturning strength (i.e., AMOC, $M_{SO}$, and $M_{WMT}$).

### 3.1 Global controls on OHUE

We first establish how the two upper ocean properties are related to OHUE in the global mean (Fig. 1a–b). Preindustrial global mean upper ocean stratification is not significantly correlated with OHUE at the $p = 0.05$ level across our ensemble of 28 CMIP6 models (Fig. 1a). In contrast, preindustrial global mean MLD is positively correlated with OHUE with a linear

correlation coefficient of $r = 0.56$ (Fig. 1b), i.e., models with a deeper global mean mixed layer tend to have a higher OHUE.

Turning now to the three overturning strength metrics (Fig. 1c–e), preindustrial AMOC strength is positively correlated across models with OHUE (Fig. 1c, $r = 0.61$). This is consistent with previous findings, but we obtain a smaller correlation coefficient for our ensemble of 28 CMIP6 models than for the mixed model ensemble of Gregory et al. (2023) which included 19 CMIP5 models and 14 CMIP6 models (their $r = 0.81$). A slightly stronger relationship is found for the Southern Ocean

upper cell (Fig. 1d): $M_{SO}$ and OHUE are also positively correlated ($r = 0.64$). The model MRI-ESM2-0 is an outlier with high OHUE but only moderate $M_{SO}$, removing this model from the linear fit results in a correlation of $r = 0.86$. As an alternative to the overturning metric $M_{SO}$ computed in latitude–density coordinates, we also consider the Southern Ocean overturning strength inferred from surface buoyancy fluxes, $M_{WMT}$ (Fig. 1e). This metric is not significantly correlated with OHUE at the $p = 0.05$ level in our model ensemble ($r = 0.39$, $p = 0.08$).

### 3.2 Local upper ocean controls on OHUE

The fact that global mean upper ocean stratification is not significantly correlated with OHUE across models may at first sight appear to contradict previous findings highlighting the importance of stratification for OHUE (Liu et al., 2023; Newsom et al.,







**Figure 1.** Proposed controls on ocean heat uptake efficiency (OHUE). Scatter plot between OHUE and **(a)** preindustrial global mean upper ocean (0–1500 m) stratification ($N^2$), **(b)** preindustrial global mean mixed layer depth (MLD), **(c)** preindustrial mean AMOC strength, **(d)** Southern Ocean upper cell strength, and **(e)** Southern Ocean surface buoyancy flux inferred overturning. In panels (c)–(e), only a subset of models is included due to output availability (see Table A1).

2023; Bourgeois et al., 2022). This is because globally averaged stratification or MLD are relatively crude bulk measures of the simulated upper ocean state. We now therefore extend this analysis to the local level by considering inter-model correlations between global OHUE and the two upper ocean variables at each model grid cell (Fig. 2).






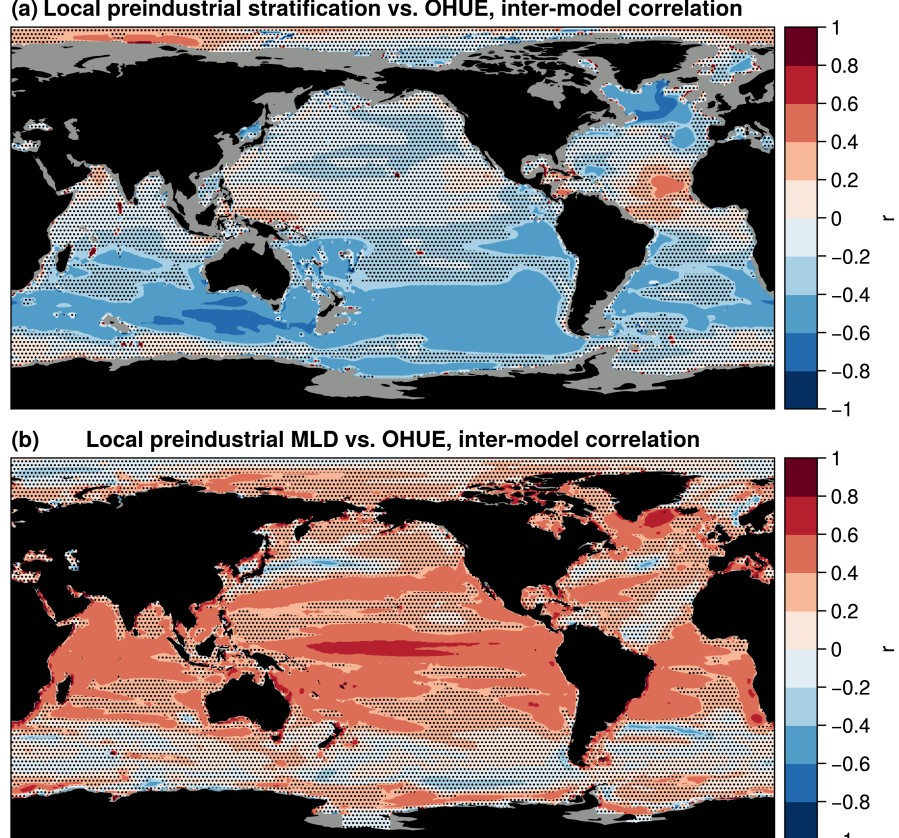

**Figure 2.** Local upper-ocean controls on ocean heat uptake efficiency (OHUE). Maps of inter-model Pearson correlation coefficient across 28 CMIP6 models between OHUE and local preindustrial annual mean **(a)** upper ocean (0–1500 m) stratification and **(b)** mixed layer depth. Stippling indicates region where the least squares linear regression slope is not significantly different from zero ($p \geq 0.05$, Wald test with $t$-distribution). In panel (a), regions where the bathymetry is less than 1500 m deep are shaded in grey.

Figure 2a shows the inter-model correlation coefficient between OHUE and local preindustrial annual mean upper ocean (0–1500 m) stratification. Unlike global average stratification (Fig. 2a), local stratification is significantly anticorrelated with OHUE in several locations. Significant correlations ($p < 0.05$) are found in two primary regions: the subpolar North Atlantic and the mid-latitude Southern Ocean. In both regions, the correlation is negative, indicating that models with greater (more stable) preindustrial stratification in these regions have a lower OHUE. In the Southern Ocean, significant negative correlations are found particularly in the Pacific and Indian sectors, whereas the signal in the southern Atlantic Ocean is less widespread. This zonally asymmetric pattern is consistent with the geography of Subantarctic Mode Water formation (McCartney, 1979; Hanawa and Talley, 2001) and subduction (Sallée et al., 2010). Apart from these two regions, a smaller patch of significant negative correlations is found in the eastern tropical Pacific. These patterns are partly dependent on the choice of the depth range over which the squared buoyancy frequency $N^2$ is integrated (Fig. A1). The negative correlation in the subpolar North





Atlantic is present for all depth choices from 0–400 m to 0–2500 m, but the negative correlation in the mid-latitude Southern Ocean is absent for 0–400 m stratification and only emerges gradually for 0–1500 m and deeper depth ranges. This suggests that the aspect of subpolar North Atlantic stratification that is important for AMOC strength is already set in the top 400 meters (i.e., the surface ocean mixed layer), while in the Southern Ocean, almost the entire water column matters for the large scale

overturning there. The decomposition of stratification into its temperature and salinity contributions (Eq. 2) shows that the subpolar North Atlantic control on OHUE is due to salinity stratification, whereas temperature stratification in this region is positively correlated with OHUE (Fig. A1). In the Southern Ocean, both temperature and salinity contribute to the negative correlation with OHUE (Fig. A1), and only their combination to total stratification results in the broad-scale signal found across the Southern Ocean in Fig. 2a.

An analogous analysis for local preindustrial annual mean MLD is shown in Figure 2b. Significant positive correlations are found in the subpolar North Atlantic as well as at low latitudes in all ocean basins; higher OHUE is thus associated with deeper mixed layers in these regions. However, in contrast to stratification, there are no significant correlations between MLD and OHUE in the mid-latitude Southern Ocean.

## 4   Upper ocean controls on meridional overturning

In the previous section, we have found significant inter-model correlations with OHUE not only for meridional overturning metrics (Fig. 1c,d), but also for regional upper ocean properties (Fig. 2). It is therefore worthwhile to investigate the potential linkages between these two categories of variables across the model ensemble, i.e. between stratification and MLD on the one hand, and overturning metrics on the other hand, as shown in Figure 3.

The left column of Figure 3 shows the inter-model correlations between local preindustrial mean upper ocean stratification
and preindustrial AMOC, $M_{SO}$ or $M_{WMT}$. Preindustrial AMOC strength is anticorrelated with subpolar North Atlantic total stratification, and weakly positively correlated with total stratification in the western Pacific (Fig. 3a). While the signal in the western Pacific is unclear and due to both temperature and salinity stratification, the negative correlation in the subpolar North Atlantic can be attributed to salinity stratification, since the temperature contribution is of the opposite sign (Fig. A2b–c).

The Southern Ocean upper cell strength, $M_{SO}$, computed in latitude–density coordinates is anticorrelated with total strati-
fication mostly in the Southern Ocean at the latitudes of the Antarctic Circumpolar Current (ACC; Fig. 3c). This can mostly be attributed to temperature stratification (Fig. A2e), which has significant negative correlations extending up to subtropical latitudes in the Pacific and Indian oceans.

The Southern Ocean upper cell strength $M_{WMT}$ inferred from surface buoyancy fluxes is also negatively correlated with total stratification in the Southern Ocean, and its correlations are higher and extend over a greater surface area (Fig. 3e) than for
the upper cell computed in latitude–density coordinates. However, for this metric, the inter-model link to stratification can be attributed solely to salinity stratification (Fig. A2i), while temperature stratification shows no significant correlation to $M_{WMT}$ in any of the major ocean basins (Fig. A2h). This is consistent with the regional hydrography, since the stratification in this region is mostly representative of the density difference between the surface ocean and the circumpolar deep water (CDW)



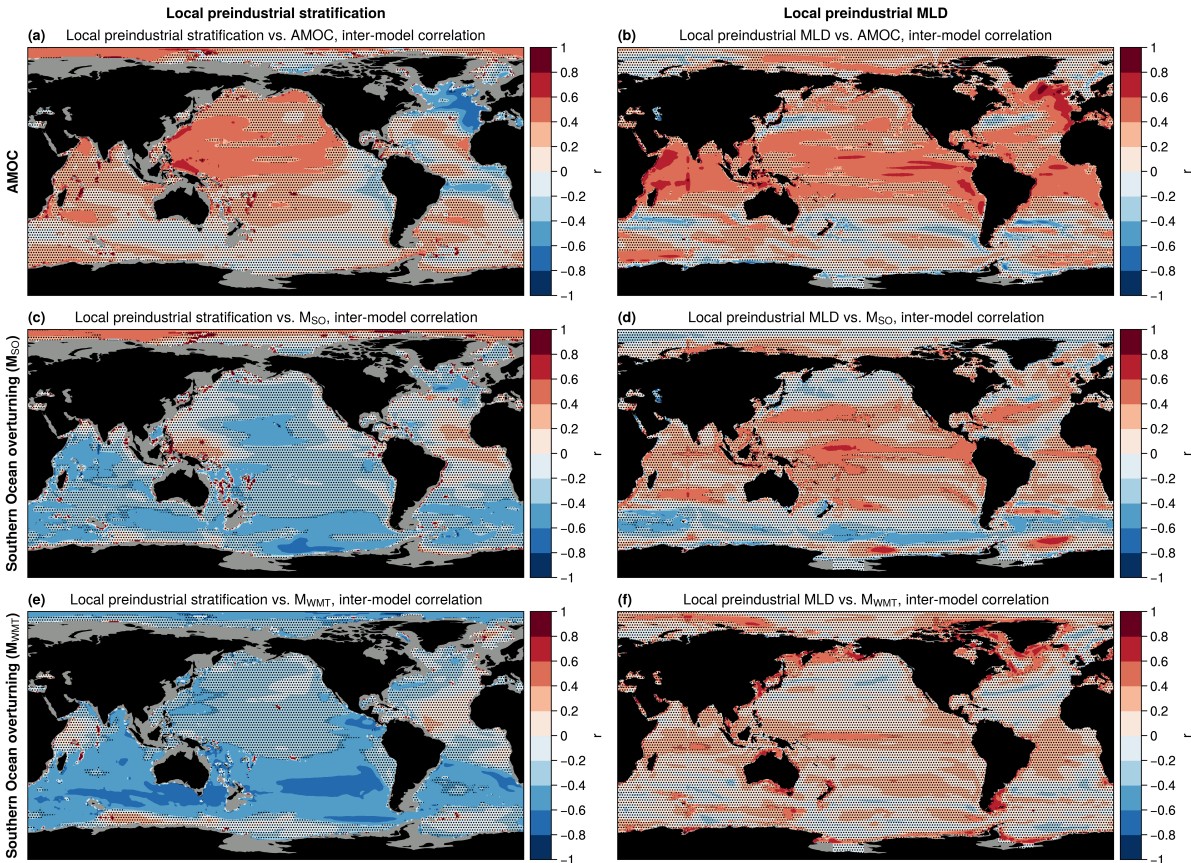

**Figure 3.** Local upper-ocean controls on meridional overturning strength in CMIP6. Left column: maps of inter-model Pearson correlation coefficient across 28 CMIP6 models between local preindustrial annual mean upper ocean (0–1500 m) stratification and **(a)** preindustrial mean AMOC strength, **(c)** Southern Ocean upper cell strength, and **(e)** Southern Ocean surface buoyancy flux inferred overturning. Right column [**(b)**, **(d)**, **(f)**]: as left column, but for local preindustrial annual mean mixed layer depth.

below, and since the conversion of CDW into lighter water is mostly due to surface freshwater fluxes (Abernathey et al., 2016;
Pellichero et al., 2018) and is qualitatively given here by $M_{\mathrm{WMT}}$.

We now turn to the links between these overturning strength metrics and local preindustrial mean MLD, shown in the right column of Figure 3. AMOC strength is positively correlated with MLD in the subpolar North Atlantic as well as at tropical latitudes in all ocean basins. This closely resembles the pattern found for the MLD–OHUE link in Figure 2b, which is a point to which we will return in the conclusions (Sect. 6).

For the two Southern Ocean overturning metrics $M_{\mathrm{SO}}$ and $M_{\mathrm{WMT}}$, a potential link to MLD is overall much less clear than for the AMOC. While $M_{\mathrm{SO}}$ is positively correlated with MLD in some regions in the tropical and subtropical Pacific, it is negatively correlated with MLD along the Polar Front in the Southern Ocean. Furthermore, the Southern Ocean overturning metric inferred from surface buoyancy fluxes, $M_{\mathrm{WMT}}$, exhibits no large-scale regions of significant correlations with MLD.




It is possible that links between the Southern Ocean overturning circulation and local MLD in the CMIP6 ensemble are more

difficult to identify than for AMOC in the North Atlantic, since Southern Ocean water mass formation and subduction locations
vary across models (Sallée et al., 2013a, b).

## 5  Stratification model bias and inter-model spread

### 5.1  Ensemble mean stratification and bias relative to observations

Although we found global mean stratification to be unrelated to OHUE (Fig. 1), there are significant links between regional

stratification and OHUE in the subpolar North Atlantic and the mid-latitude Southern Ocean (Fig. 2a). In addition, stratification
in each of these two regions is in turn related to the AMOC and Southern Ocean overturning, respectively (Fig. 4). Potential
model biases in these regions would thus have direct implications for OHUE. Beyond the foregoing analysis of inter-model
relationships between variables, it is thus insightful to investigate also the mean state, inter-model spread, and bias relative to
observations of simulated upper ocean stratification; this is shown in Figure 4.

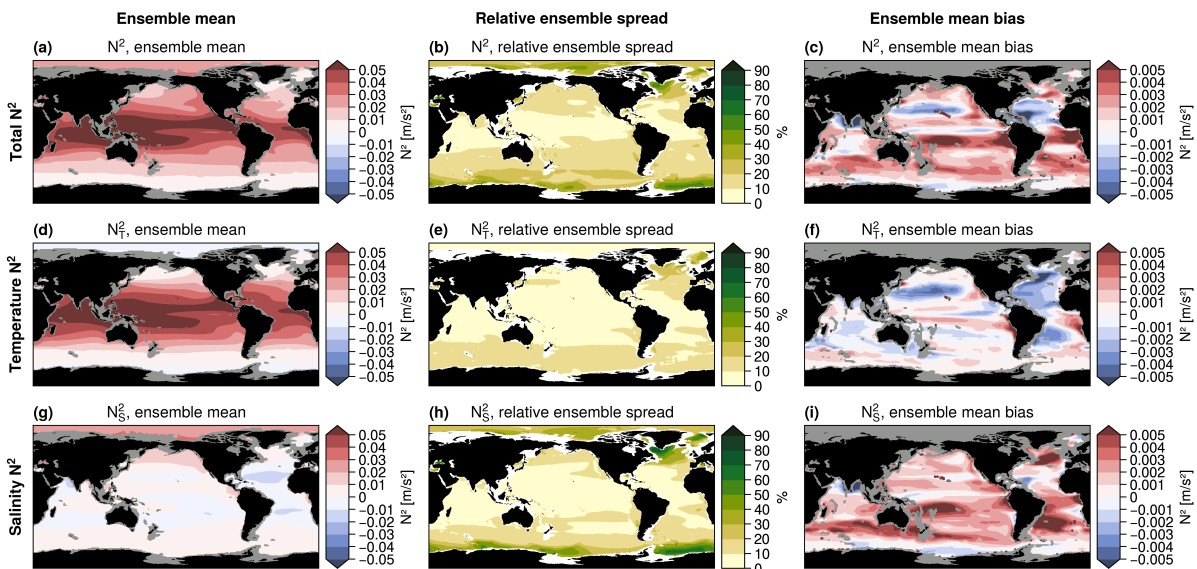

**Figure 4.** Ensemble mean stratification and bias relative to observations. **(a)** CMIP6 ensemble mean total historical stratification integrated over the 0–1500 m depth range. **(b)**, Inter-model coefficient of variation (ratio of ensemble standard deviation to ensemble mean) of total stratification. **(c)**, Bias in total stratification between CMIP6 ensemble mean and the ECCO state estimate. **(d)–(f)**, As (a)–(c) but for temperature stratification. **(g)–(i)**, As (a)–(c) but for salinity stratification. For both the model ensemble and the state estimate, stratification is averaged over the historical period 1992–2017.

The ensemble mean total stratification (Fig. 4a) has a distinct equator-to-pole gradient, with a highly stratified water column
in the tropics, and lowest stratification in the Southern Ocean and subpolar North Atlantic. Consequently, the largest relative





inter-model spread in total stratification (Fig. 4b) is found in regions with low stratification commonly associated with deep convection: the Weddell and Ross Seas in the Southern Hemisphere and the subpolar North Atlantic and Nordic Seas in the Northern Hemisphere, where the inter-model standard deviation is larger than 50% of the ensemble mean. Compared to the ECCO state estimate, the CMIP6 ensemble is too stratified over most of the ocean (Fig. 4c), especially in the equatorial Pacific and Atlantic, where the bias reaches values of up to 10% of the ensemble mean, and in the mid-latitude Southern Ocean.

The temperature contribution to stratification dominates the magnitude and pattern of the ensemble mean total stratification in the low-to-mid latitudes (Fig. 4d), while the mean salinity contribution is responsible for stabilizing the high latitude oceans (Fig. 4g). This is a consequence of the nonlinear equation of state for seawater which diminishes the influence of temperature on density in cold water (Roquet et al., 2015). Relative to the average total stratification, there is a larger inter-model spread in salinity stratification than in temperature stratification (Fig. 4e,h), especially in the high-latitude Southern Ocean around Antarctica and in the North Atlantic subpolar gyre and Nordic Seas. Despite its subordinate role in setting the mean global stratification, the salinity contribution is thus a deciding factor in the inter-model spread in total stratification. Furthermore, salinity stratification also dominates the model bias relative to the state estimate (Fig. 4i), with relatively large positive salinity stratification biases in the Southern Ocean and subpolar North Atlantic, while temperature stratification biases are small in magnitude except for a negative bias in the Atlantic basin (Fig. 4f). It should be recalled that the biases documented here are those of the CMIP6 ensemble mean; individual model biases may differ.

## 5.2 Regional coherence of stratification inter-model links

The fact that OHUE is unrelated to global mean stratification (Fig. 1a) and instead sensitive to stratification in disconnected regions of both the Northern and Southern Hemispheres (Fig. 2a) which additionally exhibit common biases relative to observations (Fig. 4) motivates a closer analysis of the inter-model spread in regional stratification patterns.

An inter-model empirical orthogonal function (EOF) analysis on the model ensemble's preindustrial annual mean stratification patterns reveals two principal modes of inter-model spread (Fig. 5), which together explain 55% of the inter-model variance (the third leading mode explains only 5.6% of the variance). The first EOF (Fig. 5a) explains 39% of the inter-model variance and consists of a broadly uniform large-scale coherence including the Pacific and Indian ocean basins and the Southern Ocean, but with no signal in the North Atlantic. This means that, to first order, model biases in preindustrial stratification in the Pacific, Indian, and Southern oceans tend to covary across models, whereas the North Atlantic stratification varies independently. The first-order independence of North Atlantic stratification from other regions can also be seen from an unsupervised classification of vertical stratification profiles (Fig. A7), where the North Atlantic is associated with a stratification profile not found in any other ocean basin or in the Southern Hemisphere. The same pattern as in the first EOF can be seen by considering the preindustrial inter-model correlation of local stratification with global mean stratification (Fig. 5b). Global mean stratification is correlated with local stratification across the Pacific, Indian, and Southern oceans, but not in the North Atlantic. This shows that the principal component associated with the first EOF (Fig. 5a) is strongly correlated to the global mean stratification (Fig. A3).



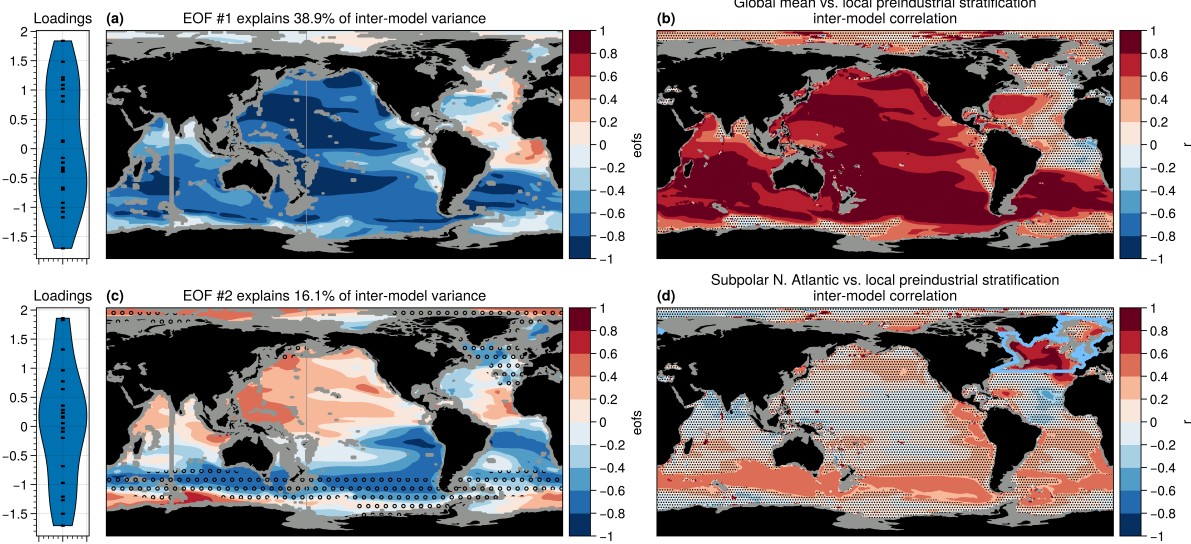

**Figure 5.** Regional coherence of inter-model stratification spread. Panels **(a)** and **(c)** show respectively the first and second mode of inter-model EOF analysis on preindustrial annual mean upper ocean stratification (see Methods). The violin plots in panels (a) and (c) show the ensemble distribution of the normalized loadings for each EOF. In panel (c), stippling indicates areas with surface density in the range $25.75 \text{ kg m}^{-3} < \sigma_0 < 27 \text{ kg m}^{-3}$. Panels **(b)** and **(d)** show the inter-model correlation between the local preindustrial stratification and either **(b)** the global mean preindustrial stratification or **(d)** the subpolar North Atlantic mean preindustrial stratification. The subpolar North Atlantic region used in panel (d) is indicated by the blue contour.

The second EOF (Fig. 5c) explains 16% of the inter-model variance in preindustrial stratification. It mainly consists of a coherence including the mid-latitude Southern Ocean, subpolar North Atlantic, and eastern tropical Pacific, and a signal of opposite sign in the western tropical Pacific. This suggests that, to second order, preindustrial stratification model biases in the Southern Ocean and subpolar North Atlantic tend to be linked. Although these two regions are geographically far apart, they are physically connected by the outcropping of the same isopycnals in the range $25.75 \text{ kg m}^{-3} < \sigma_0 < 27 \text{ kg m}^{-3}$, as indicated by the stippling of sea surface density in Figure 5c. This link is further illustrated by the inter-model correlation of local stratification with stratification averaged over the subpolar North Atlantic (indicated by the contour in Figure 5d). Apart from a trivial positive correlation in the subpolar North Atlantic itself, we find a circumpolar band of positive inter-model correlation in the mid-latitude Southern Ocean.

Further EOF modes are not explored in detail here since they each explain less than 6% of the inter-model variance. Still, the three following EOFs all have a signal of the same sign in the Southern Ocean and subpolar North Atlantic (Fig. A4), strengthening the inter-model link between these regions found in the second EOF.

The distinct role of temperature and salinity stratification at setting these patterns of inter-model spread can be seen by applying the EOF analysis to temperature and salinity stratification separately (Fig. A5). It is apparent that the first two inter-





model EOFs in total stratification (Fig. 5a,c) resemble the first two EOFs of salinity stratification (Fig. A5b,d). In contrast,
the first EOF of temperature stratification (Fig. A5a) consists of a broad low- to mid-latitude pattern including the North
Atlantic, and the second EOF (Fig. A5c) shows an approximate hemispheric dipole signal with opposite sign between the
Southern Ocean and the Northern Hemisphere oceans. This implies that inter-model spread in patterns of salinity stratification
are decisive for setting the patterns of total stratification, which in turn control OHUE (Fig. 2a).

It is furthermore interesting to note that temperature and salinity stratification ($N_T^2$ and $N_S^2$) do not vary independently across
the model ensemble: inter-model biases in temperature and salinity stratification tend to compensate each other in the high-
latitude Southern Ocean and in the North Atlantic, meaning that models with strong salinity stratification tend to have weak
temperature stratification at these locations, and vice versa (Fig. A6a). In addition, a difference in total stratification between
two models tends to coincide with a difference in salinity stratification of the same sign across almost all of the global ocean
(Fig. A6c), while temperature stratification is positively correlated with total stratification only over of the low- to mid-latitude
oceans (Fig. A6b). These findings partly explain the success of the emergent constraint by Liu et al. (2023) between sea surface
salinity as a proxy for $N_S^2$ and OHUE.

## 6  Discussion and conclusions

### 6.1  Schematic summary of principal inter-model relationships between variables controlling OHUE

The schematic in Figure 6 summarizes the inter-model relationships found in this study between local upper ocean stratification,
local mixed layer depth, various meridional overturning strength metrics, and OHUE. We now summarize our findings for the
most important connections, depicted as arrows and labelled with lowercase letters in Figure  6.

### a) Subpolar North Atlantic stratification ($N_{\text{N.Atl.}}^2$) and Southern Ocean stratification ($N_{\text{SO}}^2$)

We have identified a coherent pattern of inter-model spread in preindustrial stratification linking the subpolar North Atlantic and
the mid-latitude Southern Ocean (Fig. 5c,d). Although this mode of inter-model variability explains only 16% of inter-model
variance in preindustrial stratification (compared to 39% for the leading mode), it is key for driving differences in OHUE
between models. Indeed, the loadings of this second EOF are correlated with OHUE across the model ensemble (Pearson
$r = 0.57$, $p < 0.05$). This pattern of North Atlantic–Southern Ocean coherence is also found in the inter-model correlation
between total preindustrial stratification and OHUE (Fig. 2a), and in the ensemble mean bias of historical total and salinity
stratification with respect to observations (Fig. 4c,i).
The physical link between stratification in the mid-latitude Southern Ocean and the subpolar North Atlantic is illustrated by
the outcropping of the same isopycnals in these two regions (Fig. 5c). In both regions, permanent stratification is dominated by
the internal pycnocline of the global ocean, which separates the shallow northward and deep southward limbs of the AMOC
(Gnanadesikan, 1999; Klocker et al., 2023). An inter-hemispheric connection via the AMOC has also been shown to explain



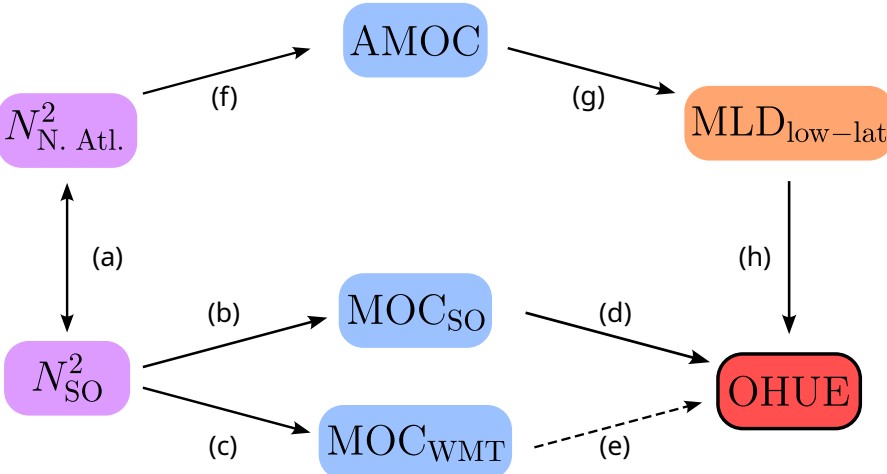

**Figure 6.** Schematic illustrating the inter-model links between key ocean properties. Arrows indicate the identified physically-based inter-model relationships, and the dashed arrow labelled (e) indicates the unclear relationship between $M_{\mathrm{WMT}}$ and OHUE.

common temperature biases in the Southern Ocean of CMIP6 models (Luo et al., 2023). Certain characteristics of the subpolar
North Atlantic can thus be proxies for those of the Southern Ocean and vice versa.

**b) Southern Ocean stratification ($N_{\mathrm{SO}}^2$) and upper cell strength ($\mathrm{MOC_{SO}}$)**

Southern Ocean stratification impacts the strength of the Southern Ocean upper overturning cell $M_{\mathrm{SO}}$ computed in latitude–
density coordinates (Fig. 3c). However, this correlation is relatively weak ($r < 0.6$ at most locations) and its spatial pattern is
rather discontinuous, although consistent with the documented regions of water-mass formation feeding the upper overturning
cell (east Indian and east Pacific basins in the latitude range 40°S–60°S; e.g., Sallée et al., 2010).

**c) Southern Ocean stratification ($N_{\mathrm{SO}}^2$) and upper cell strength inferred from surface buoyancy fluxes ($\mathrm{MOC_{WMT}}$)**

The upper cell strength inferred from surface buoyancy fluxes, $M_{\mathrm{WMT}}$, was used as an alternative measure of Southern Ocean
overturning. It is impacted by stratification across the Southern Ocean and from latitudes of the ACC up to the subtropics
(Fig. 3g), with higher correlations than for the alternative metric $M_{\mathrm{SO}}$.

**d) Upper cell strength ($\mathrm{MOC_{SO}}$) and OHUE**

The strength of the Southern Ocean upper overturning cell $M_{\mathrm{SO}}$ computed in latitude–density coordinates is well correlated
with OHUE (Fig. 1d), and when ignoring the outlier model MRI-ESM2-0, the correlation coefficient ($r = 0.86$) is much higher
than that for AMOC ($r = 0.61$). (In a different model ensemble, Gregory et al. (2023) found a correlation coefficient between
AMOC and OHUE of $r = 0.83$.)





**e) Upper cell strength inferred from surface buoyancy fluxes ($\text{MOC}_{\text{WMT}}$) and OHUE**

The upper cell strength inferred from surface buoyancy fluxes, $M_{\text{WMT}}$, was found to be not significantly correlated with OHUE ($r = 0.39$, $p = 0.08$).

**f) Subpolar North Atlantic stratification ($N^2_{\text{N.Atl.}}$) and AMOC**

Preindustrial upper ocean stratification in the subpolar North Atlantic is anticorrelated with preindustrial AMOC strength (Fig. 3a). This is consistent with theoretical understanding and modeling results from previous studies which have shown that AMOC strength in CMIP6 is influenced by North Atlantic stratification (Nayak et al., 2024), especially in the Labrador Sea and due to salinity stratification (Jackson et al., 2023; Lin et al., 2023; Jackson and Petit, 2023). This is because stratification in this region inhibits the formation of North Atlantic Deep Water which feeds the southward branch of the AMOC, mostly via open ocean deep convection in these models (Heuzé, 2021).

**g) AMOC and low-latitude mixed layer depth ($\text{MLD}_{\text{low-lat}}$)**

Preindustrial AMOC strength is positively correlated with preindustrial MLD in the subpolar North Atlantic as well as in the low latitudes in all ocean basins (Fig. 3b). Subpolar North Atlantic MLD is a proxy for deep convection (Jackson and Petit, 2023; Heuzé, 2021), and its connection to AMOC strength is consistent with process understanding and related to point f) above (Jackson et al., 2023).

However, the reason for the link between AMOC and low-latitude mixed layer depths is unclear. Since significant positive correlations are not only found in the Atlantic, but also extend across the Pacific and Indian basins, it is possible that this relationship is not directly caused by a physical mechanism, but rather due to the spatial coherence of inter-model MLD spread, analogous to stratification in Section 5.2. Indeed, an inter-model EOF analysis applied to preindustrial annual mean MLD reveals a first-order coherence between subpolar North Atlantic MLD and global MLDs including the tropics (Fig. A8a), with the second- and third-order EOFs respectively containing the variance in the high and low latitudes separately (Fig. A8b–c).

**h) Low–latitude mixed layer depth ($\text{MLD}_{\text{low-lat}}$) and OHUE**

Preindustrial mixed layer depth in the low latitudes is positively correlated to OHUE (Fig. 2b). One hypothesis to explain this is that the mixed layer depth at these latitudes quantifies the thermal capacity of the ocean, since most of the radiative forcing is applied to the ocean surface at these latitudes (Gregory et al., 2023) and deeper mixed layers have a higher heat capacity. Furthermore, since sea surface temperatures are high and vertical temperature gradients are strong in the low latitudes, the modeled mixed layer depth there may be sensitive to the parameterization of vertical mixing of heat in these models. The representation of this mixing also impacts OHUE (Newsom et al., 2023), possibly contributing to the link between low-latitude MLD and OHUE.



## 6.2 Synthesis

We are now in a position to answer the questions posed in the Introduction of this study.

### 6.2.1 In which oceanic regions does stratification control OHUE?

The key regions where preindustrial stratification controls OHUE are the subpolar North Atlantic and the mid-latitude Southern Ocean (Fig. 2a). These two regions are linked together via the second-order mode of inter-model stratification spread (Fig. 5), and they are precisely the regions where ensemble mean historical stratification is biased high (Fig. 4c) due to biased salinity stratification (Fig. 4i). This is consistent with the findings of Liu et al. (2023) who showed that CMIP6 models tend to overestimate salinity stratification, particularly in these regions (their Figure 3a), and that salinity stratification approximated via sea surface salinity can be used to constrain OHUE. Our results demonstrate that it is possible that only the Southern Ocean stratification has a direct effect on OHUE through its influence on the large scale overturning circulation (Fig. 3d–i); the subpolar North Atlantic stratification could be anticorrelated with OHUE due to its connection with Southern Ocean stratification (Fig. 5) rather than due to a direct influence on OHUE. This would be consistent with previous findings showing that the actual amount of anomalous heat entering the North Atlantic and being subducted by the AMOC is small compared to the OHU occurring in the mid-latitude Southern Ocean (Frölicher et al., 2015; Cheng et al., 2022), and that changes in the strength of OHUE and AMOC under transient forcing are uncorrelated (Stolpe et al., 2018). The direct link between OHUE and Southern Ocean stratification, rather than North Atlantic stratification, is further illustrated by a comparison of the upper ocean stratification definition used here with the pycnocline depth index defined by Newsom et al. (2023) (Fig. A9). This near-global (60°S–60°N) pycnocline depth index has been shown to nicely constrain OHUE (Newsom et al., 2023), and we show here that it is strongly anticorrelated with local stratification in the Southern Ocean but not in the subpolar North Atlantic (Fig. A9a).

### 6.2.2 How do biases in temperature and salinity stratification differ in their control on OHUE?

Salinity stratification biases in CMIP6 have a dominant role for OHUE due to several reasons. First, the inter-model spread in total stratification in key regions is dominated by spread in salinity stratification (Fig. 4h). Second, salinity stratification sets the spatial patterns of inter-model stratification spread as determined by the inter-model EOF analysis (Figs. 5 and A5). Finally, the pattern of the bias of CMIP6 ensemble mean stratification with respect to the ECCO state estimate is driven by the bias in salinity stratification (Fig. 4c,i). This is consistent with the dominant role of salinity stratification for OHUE found by Liu et al. (2023). However, temperature stratification also plays a role, in particular for setting the mean strength of global total stratification.

### 6.2.3 What explains the positive correlation between AMOC strength and OHUE across CMIP6 models?

AMOC strength is directly controlled by subpolar North Atlantic stratification. The positive correlation of AMOC with OHUE can be explained by two factors: i) North Atlantic stratification is connected to Southern Ocean stratification physically via the internal pycnocline (separating shallow northward and deep southward limbs of the global overturning) and statistically via the



second EOF of inter-model stratification spread. We argue that Southern Ocean stratification influences in turn OHUE via the overturning circulation; ii) both AMOC and OHUE are related to low-latitude MLD as a proxy of thermal capacity.

These two factors represent the upper and lower branches connecting AMOC to OHUE in the schematic in Figure 6, and presumably they both contribute to the positive correlation between AMOC and OHUE. Our analysis thus supports the hypoth-
esis that the AMOC is not the mechanism actively controlling OHUE (Gregory et al., 2023). This hypothesis concurs with the observation that the amount of heat entering the North Atlantic and being subducted by the AMOC is relatively small compared to Southern Ocean OHU (Frölicher et al., 2015; Cheng et al., 2022), due to aerosol-induced cooling in the North Atlantic and larger subduction rates in the Southern Ocean (Williams et al., 2024).

### 6.2.4 What is the role of meridional overturning in the Southern Ocean for OHUE?

Our results indicate that the AMOC might not be the ocean circulation directly affecting OHUE by transporting heat into the ocean interior, and that, instead, it is the Southern Ocean upper overturning cell which has a direct impact on OHUE. However, the link between Southern Ocean stratification to OHUE via Southern Ocean overturning is difficult to quantify. The connection between Southern Ocean stratification and Southern Ocean overturning is clearest when using an overturning metric inferred from surface buoyancy fluxes ($M_{\mathrm{WMT}}$, Fig. 3c,e), but the link from Southern Ocean overturning to OHUE is only significant
when using an overturning metric calculated directly from meridional velocities in latitude–density coordinates (Fig. 1d,e). The two Southern Ocean overturning metrics $M_{\mathrm{SO}}$ and $M_{\mathrm{WMT}}$ are uncorrelated across the model ensemble and have distinct advantages and disadvantages. Although $M_{\mathrm{SO}}$ directly quantifies the strength of the upper overturning cell actively transporting heat into the ocean interior, it is not a perfect measure of subduction across the Southern Ocean (Sallée et al., 2012). Indeed subduction occurs at different latitudes and densities around the Southern Ocean, and across members of the CMIP6 ensemble,
such that it is difficult to obtain an accurate measure of Southern Ocean subduction rates from CMIP6 output. The $M_{\mathrm{WMT}}$ metric instead quantifies the total upwelling in the Southern Ocean via surface buoyancy fluxes but does not include the effect of mixing, which plays an important role in the Southern Ocean overturning circulation (Sallée et al., 2013b; Evans et al., 2018).

The foregoing discussion highlights the practical difficulties in quantifying Southern Ocean vertical transports in a large
multi-model ensemble. By contrast, subduction in the subpolar North Atlantic is more straightforward to quantify via the AMOC streamfunction, and this partly explains the relative success of AMOC strength as a metric to quantify ocean overturning rates in models and to correlate with climate metrics such as OHUE (Kostov et al., 2014; Gregory et al., 2023). More detailed output variables in future model intercomparisons allowing to characterize regional subduction or ventilation rates would be instrumental to better pin down physical controls of ocean heat and carbon uptake.

*Code and data availability.* All model output and observational data used in this study are freely available. CMIP6 model output is available from the Earth System Grid Federation at https://esgf-node.llnl.gov/projects/cmip6/. Data from the ECCO state estimate are available at https://www.ecco-group.org/products-ECCO-V4r4.htm.



The processed data and Python code used to produce the figures in this study are available at *<Zenodo URL to be provided upon acceptance>*

**Appendix A**

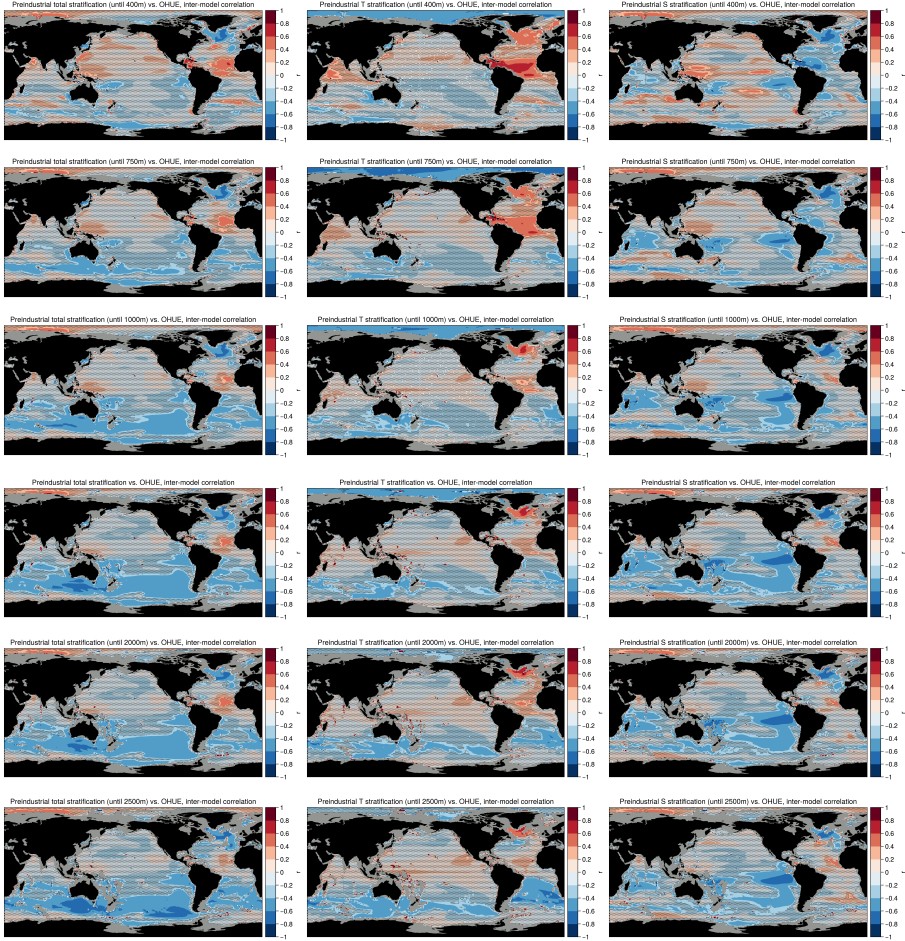

**Figure A1.** Maps of inter-model Pearson correlation coefficient between OHUE and local preindustrial annual mean total (left column), temperature (middle column), and total (right column) stratification across 28 CMIP6 models, with stratification in the depth ranges **(a)–(c)** 0–400 m, **(d)–(f)** 0–750 m, **(g)–(i)** 0–1000 m, **(j)–(l)** 0–1500 m, **(m)–(o)** 0–2000 m, and **(p)–(r)** 0–2500 m. Stippling indicates region where the linear slope is not significantly different from zero ($p \geq 0.05$, Wald test with $t$-distribution). Regions where the bathymetry is less than 1500 m deep are shaded in grey.




**Table A1.** CMIP6 models used in this study.

| Model | piControl | 1pctCO2 | Data available for AMOC/$M_{SO}$/$M_{WMT}$ | Reference |
|---|---|---|---|---|
| CanESM5 | r1i1p1f1 | r1i1p1f1 | Y/Y/Y | Swart et al. (2019b) |
| CanESM5-CanOE | r1i1p2f1 | r1i1p2f1 | Y/Y/Y | Swart et al. (2019a) |
| CMCC-CM2-SR5 | r1i1p1f1 | r1i1p1f1 | Y/Y/Y | Lovato and Peano (2020) |
| CMCC-ESM2 | r1i1p1f1 | r1i1p1f1 | Y/Y/Y | Lovato et al. (2021) |
| CNRM-CM6-1 | r1i1p1f2 | r1i1p1f2 | Y/Y/Y | Voldoire (2018) |
| CNRM-CM6-1-HR | r1i1p1f2 | r1i1p1f2 | Y/N/Y | Voldoire (2019) |
| CNRM-ESM2-1 | r1i1p1f2 | r1i1p1f2 | Y/Y/Y | Seferian (2018) |
| ACCESS-ESM1-5 | r1i1p1f1 | r1i1p1f1 | Y/Y/Y | Ziehn et al. (2019) |
| ACCESS-CM2 | r1i1p1f1 | r1i1p1f1 | Y/Y/Y | Dix et al. (2019) |
| EC-Earth3 | r1i1p1f1 | r3i1p1f1 | N/Y/N | EC-Earth Consortium (EC-Earth) (2019a) |
| EC-Earth3-CC | r1i1p1f1 | r1i1p1f1 | Y/Y/Y | EC-Earth Consortium (EC-Earth) (2020b) |
| EC-Earth3-Veg | r1i1p1f1 | r1i1p1f1 | Y/Y/N | EC-Earth Consortium (EC-Earth) (2019b) |
| EC-Earth3-Veg-LR | r1i1p1f1 | r1i1p1f1 | N/Y/N | EC-Earth Consortium (EC-Earth) (2020a) |
| IPSL-CM6A-LR | r1i1p1f1 | r1i1p1f1 | Y/Y/Y | Boucher et al. (2018) |
| MIROC6 | r1i1p1f1 | r1i1p1f1 | Y/Y/Y | Tatebe and Watanabe (2018) |
| HadGEM3-GC31-LL | r1i1p1f1 | r1i1p1f3 | Y/Y/Y | Ridley et al. (2018) |
| HadGEM3-GC31-MM | r1i1p1f1 | r1i1p1f3 | Y/N/Y | Ridley et al. (2019) |
| UKESM1-0-LL | r1i1p1f2 | r1i1p1f2 | Y/Y/Y | Tang et al. (2019) |
| MPI-ESM1-2-HR | r1i1p1f1 | r1i1p1f1 | Y/Y/Y | Jungclaus et al. (2019) |
| MPI-ESM1-2-LR | r1i1p1f1 | r1i1p1f1 | Y/Y/Y | Wieners et al. (2019) |
| MRI-ESM2-0 | r1i1p1f1 | r1i1p1f1 | Y/Y/Y | Yukimoto et al. (2019) |
| GISS-E2-1-G | r1i1p1f2 | r1i1p1f1 | N/Y/N | NASA/GISS (2018) |
| CESM2 | r1i1p1f1 | r1i1p1f1 | Y/N/N | Danabasoglu (2019b) |
| CESM2-WACCM | r1i1p1f1 | r1i1p1f1 | Y/N/N | Danabasoglu (2019a) |
| NorESM2-LM | r1i1p1f1 | r1i1p1f1 | Y/Y/N | Seland et al. (2019) |
| NorESM2-MM | r1i1p1f1 | r1i1p1f1 | Y/Y/Y | Bentsen et al. (2019) |
| GFDL-CM4 | r1i1p1f1 | r1i1p1f1 | Y/N/Y | Guo et al. (2018) |
| GFDL-ESM4 | r1i1p1f1 | r1i1p1f1 | Y/N/Y | Krasting et al. (2018) |



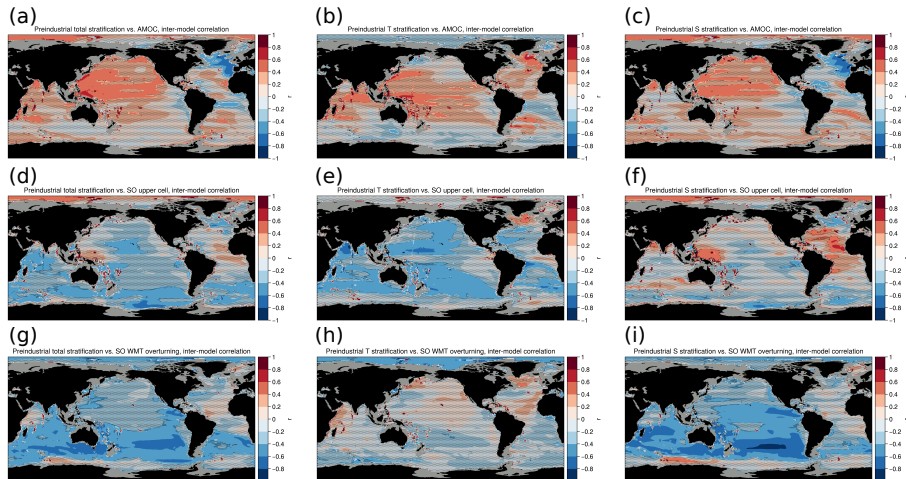

**Figure A2.** Inter-model relation between stratification and overturning cells. **(a)–(c)**: inter-model correlation between preindustrial 0–1500 m stratification and AMOC for total (left column), temperature (middle column), and salinity (right column) stratification. **(d)–(f)**: as in (a)–(c) but for the Southern Ocean upper cell in density coordinates. **(g)–(i)**: as in (a)–(c) but for the Southern Ocean overturning strength inferred from surface buoyancy fluxes (see Methods). Note that the first column of this figure is the same as the first column of Fig. 3 in the main text.

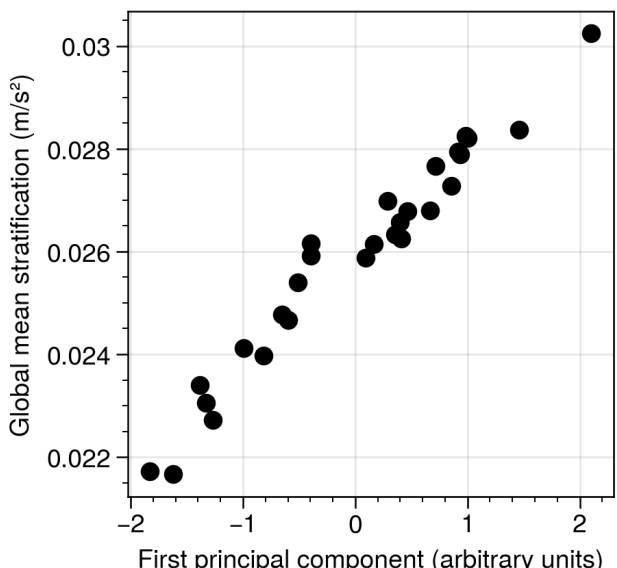

**Figure A3.** Scatter plot between the first principal component of the inter-model EOF analysis on preindustrial stratification (see Sect. 5.2) and global mean preindustrial stratification for each CMIP6 model.

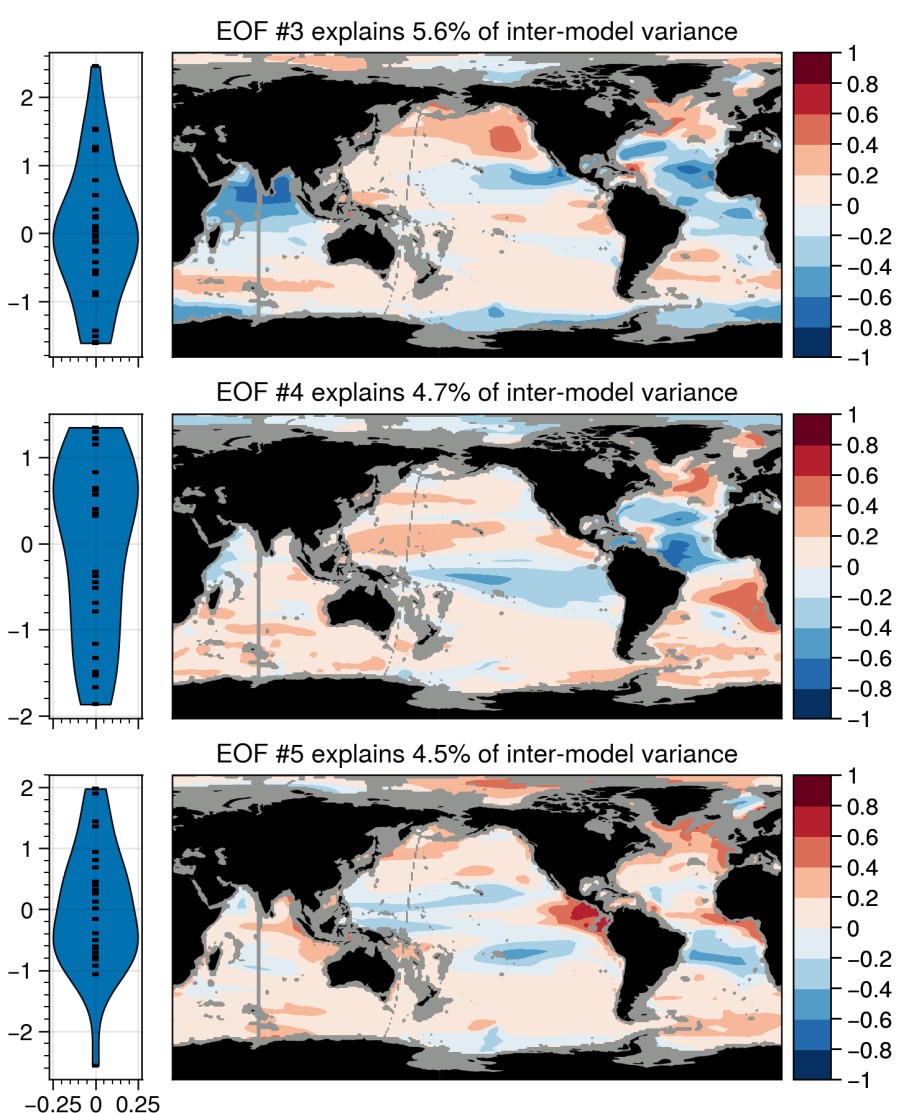

**Figure A4.** Modes 3 to 5 of inter-model empirical orthogonal function analysis on preindustrial annual mean upper ocean stratification.



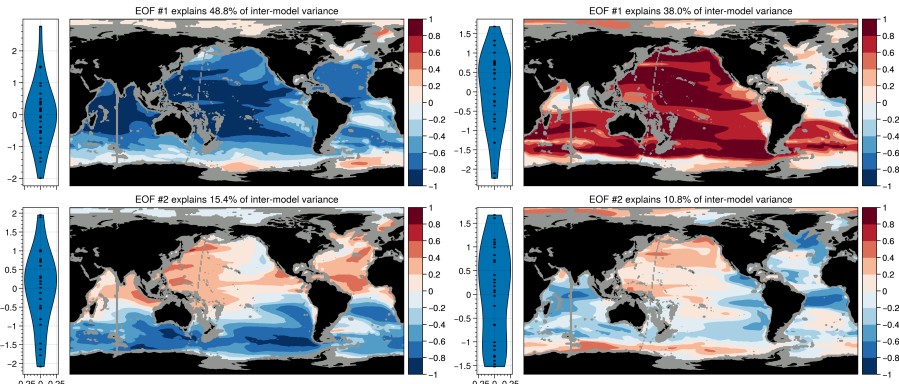

**Figure A5.** Left column: first and second mode of inter-model empirical orthogonal function analysis on preindustrial annual mean upper ocean temperature stratification. Right column: as left column, but for salinity stratification.

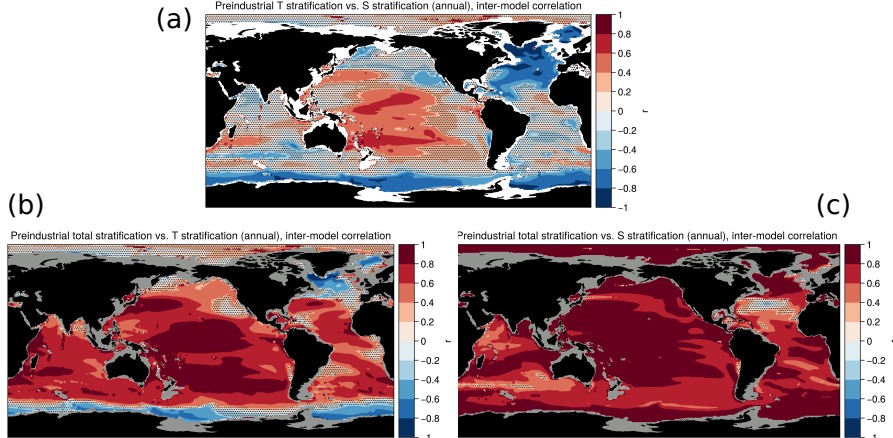

**Figure A6. (a)**, Map of inter-model correlation between preindustrial local 0–1500 m temperature stratification and salinity stratification. **(b)**, Same as (a) but between total stratification and temperature stratification. **(c)**, Same as (a) but between total stratification and salinity stratification.



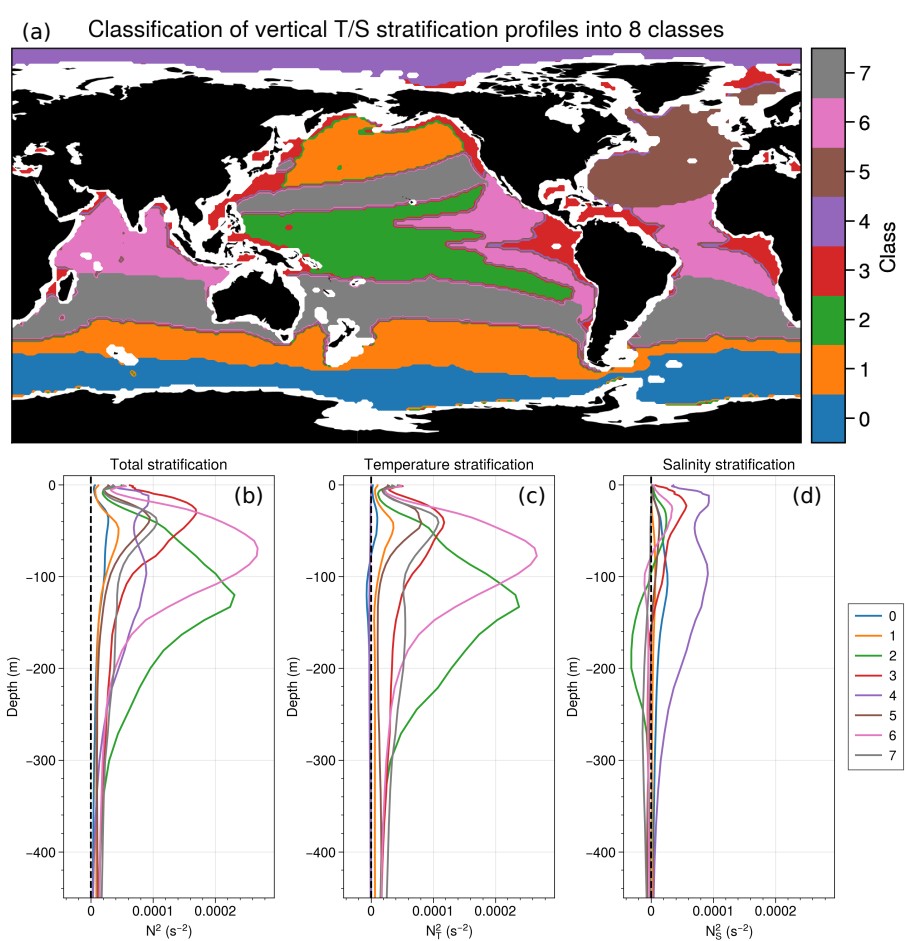

**Figure A7.** Classification of vertical stratification profiles. **(a)**, map showing the geographical location of identified classes. **(b)–(d)**, median vertical stratification profiles of each class (for total, temperature, and salinity stratification).

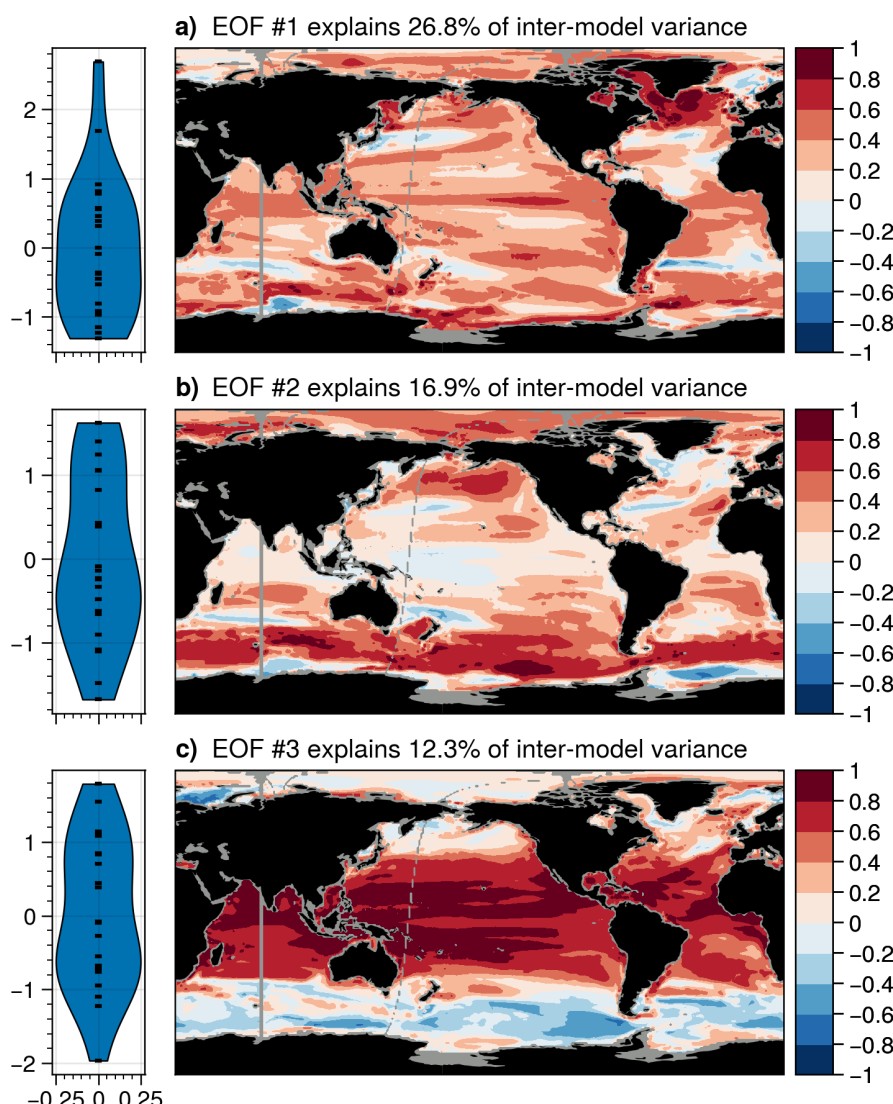

**Figure A8.** EOF analysis on preindustrial MLD. First three modes of inter-model empirical orthogonal function analysis on preindustrial annual mean MLD after removing 5 outlier models (see Methods).







**Figure A9.** Inter-model correlation across 28 CMIP6 models between the pycnocline depth metric defined by Newsom et al. (2023) and local preindustrial annual mean **(a)** total, **(b)** temperature, and **(c)** salinity stratification.





*Author contributions.* LV conceived the study, performed the data processing and analysis, and wrote the initial draft. JBS and CdL were responsible for supervision and funding acquisition. All authors contributed to interpreting the results and revising the paper.

*Competing interests.* The authors declare that they have no conflict of interest.

*Acknowledgements.* The authors thank Juliette Mignot, Ric Williams, and Robin Waldman for discussions. LV, JBS, and CdL received
funding from the European Union's Horizon 2020 research and innovation programme under grant agreement no. 821001 (SO-CHIC).



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
