# Peer review of "Stratification and overturning circulation are intertwined controls on ocean heat uptake efficiency in climate models"

_EGUsphere, 2024_

## Referee Comment (RC1)

Dear editor and authors,

The Vogt et al. manuscript offers a step-change in our understanding of the relationship between the preindustrial control state of current generation models and their ocean heat uptake under anthropogenic forcing. The manuscript is really well written. Apart from some minor clarifications, there are two important issues that I would like to see addressed.

First, I would like the authors to probe more deeply into their first principal component of intermodel variance in the preindustrial annual mean stratification. It seems to me that EOF1 is very much related to the strength of the ocean gyres and may point to intermodel differences and biases in the surface wind-stress curl.

Second, I would like the authors to clarify whether some individual CMIP models exhibit a bias towards convection in the Labrador Sea while underestimating processes in the Irminger Sea. How do such intermodel differences correlate with the AMOC in the preindustrial and forced experiments?

I therefore recommend **minor revisions**.

Major comments:

Lines 269-271 and the Synthesis section: Are you sure that "broadly uniform" is the best way to describe the first EOF of preindustrial annual mean stratification? I would argue that EOF1 is very much related to the strength of ocean gyres, with the pattern being weighted towards the stronger subtropical gyres compared to the weaker ones.
    Is EOF1 related in any way to the near-surface component of the large-scale atmospheric circulation? Does it represent systematic biases and intermodel spread in the surface wind stress curl?

Line 210: Are there some models that exhibit convection only in the Labrador Sea and others that have convection in both the Labrador and the Irminger basins in their control state? Is there also a connection between the spatial pattern of the preindustrial North Atlantic MLD and the AMOC weakening in response to forcing?

Minor comments:

Line 42: add "in a historical context" to distinguish from future scenarios

Line 148: Specify what time period of the preindustrial control run do you include in the averaging? Is it only the period preceding the branching, the period starting af the same time as the branching, or the full preindustrial control simulation?

Lines 150-151: Does this process leave you with 23 models, or you get 28 models left after the elimination of outliers?

Line 158: 28 models before or after the removal of outliers?

Lines 518-520, this manuscript is now published, and the updated reference is:
Gregory, J.M., Bloch-Johnson, J., Couldrey, M.P. et al. A new conceptual model of global ocean heat uptake. Clim Dyn 62, 1669–1713 (2024). https://doi.org/10.1007/s00382-023-06989-z

---

## Author Response (AR1)

**1 Response to reviewer 1**

**General Comments**

Dear editor and authors, The Vogt et al. manuscript offers a step-change in our understanding of the relationship between the preindustrial control state of current generation models and their ocean heat uptake under anthropogenic forcing. The manuscript is really well written. Apart from some minor clarifications, there are two important issues that I would like to see addressed.

First, I would like the authors to probe more deeply into their first principal component of intermodel variance in the preindustrial annual mean stratification. It seems to me that EOF1 is very much related to the strength of the ocean gyres and may point to intermodel differences and biases in the surface wind-stress curl.

Second, I would like the authors to clarify whether some individual CMIP models exhibit a bias towards convection in the Labrador Sea while underestimating processes in the Irminger Sea. How do such intermodel differences correlate with the AMOC in the preindustrial and forced experiments?

I therefore recommend **minor revisions**.

**Response:**

We thank the reviewer for their positive evaluation and their constructive and helpful comments, which have substantially improved our manuscript. We have taken each comment into account, provided responses to each point below, and adapted the manuscript accordingly.
* * *
**Comment 1.1**

Lines 269-271 and the Synthesis section: Are you sure that "broadly uniform" is the best way to describe the first EOF of preindustrial annual mean stratification? I would argue that EOF1 is very much related to the strength of ocean gyres, with the pattern being weighted towards the stronger subtropical gyres compared to the weaker ones. Is EOF1 related in any way to the near-surface component of the large-scale atmospheric circulation? Does it represent systematic biases and intermodel spread in the surface wind stress curl?
* * *
**Response:**

We thank the reviewer for this comment. In order to investigate this idea, we computed preindustrial mean wind stress curl at the ocean surface in each model (Fig. R1 below), and produced maps of inter-model correlation between local wind stress curl and EOF #1 (the "broadly uniform" first principal component of preindustrial stratification; Fig. R2a below) as well as global mean preindustrial stratification (Fig. R2b below). For both EOF #1 and global mean stratification, no statistically significant inter-model correlation to local wind stress curl can be found in any geographical region.

[Figure]

Figure R1: Maps of preindustrial mean wind stress curl in each climate model.

Since this investigation of wind stress curl does not yield conclusive results, we also computed the preindustrial strength of the Northern and Southern subtropical overturning cells in the Atlantic and Pacific for each model. We calculated each cell's volume transport as the maximum value of the preindustrial latitude-depth overturning streamfunction (also used to calculate AMOC strength in the main manuscript) in the depth range 0–200m and the latitude range 0–35°N/S. Figure R3 below shows scatter plots between the loadings of the stratification EOF #1 and each subtropical cell's strength. EOF #1 is anticorrelated with subtropical overturning strength in the South Atlantic as well as in both hemispheres in the Pacific, while there is no significant correlation in the North

[Figure]

Figure R2: Inter-model correlation map between local preindustrial mean wind stress curl and **(a)** the loadings of EOF #1 (first principal component) of preindustrial annual mean stratification; and **(b)** global mean preindustrial annual mean stratification.

Atlantic. The absence of a relation to the North Atlantic subtropical cell is to be expected since EOF #1 does not have a signal there (see Fig. 5a in the main manuscript). EOF #1 relates to the other three subtropical cells via local stratification.

We further test the influence of stratification on the subtropical cells by computing inter-model correlation maps between local stratification and subtropical cell strength (see Fig. R4 below). For all subtropical cells except in the North Atlantic, regions of significant anti-correlation between stratification and subtropical cell strength are apparent. However, these regions do not exactly conform to the locations of the subtropical cells in question, e.g., the South Atlantic subtropical cell strength is anticorrelated to stratification across the Southern Hemisphere including the Pacific, but this is to be expected given the pattern of EOFs #1 and #2 investigated in the main manuscript.

Given the relatively unclear relationship between EOF #1 and the subtropical cells, we have simply added the following sentence to the cited paragraph in the main manuscript:

"The pattern of the first EOF is potentially also linked to the strength of the subtropical gyres and subtropical overturning cells in the Pacific and Atlantic."

[Figure]

Figure R3: Scatter plots between the loadings of EOF #1 (first principal component) of preindustrial annual mean stratification and preindustrial subtropical overturning cell strengths across models.

[Figure]

Figure R4: Maps of inter-model correlation between local preindustrial stratification and preindustrial subtropical overturning cell strengths.
* * *
**Comment 1.2**

Line 210: Are there some models that exhibit convection only in the Labrador Sea and others that have convection in both the Labrador and the Irminger basins in their control state? Is there also a connection between the spatial pattern of the preindustrial North Atlantic MLD and the AMOC weakening in response to forcing?
* * *
**Response:**

We thank the reviewer for these ideas.

Regarding the first point about deep convection, Figure R5 below shows the average yearly maximum mixed layer depth in the North Atlantic over the preindustrial experiment in each model. Regions with high values of this metric can be understood to correspond to locations of wintertime deep convection (e.g., Heuzé and Liu 2024).

[Figure]

Figure R5: Average yearly maximum mixed layer depth in the preindustrial experiment of each CMIP6 model used in the study.

Figure R5 suggests that some models exhibit convection both in the Labrador Sea and the Irminger and Greenland-Iceland-Norwegian (GIN) Seas (e.g., the CMCC, CNRM, MPI, HadGEM, GISS, CESM2, NorESM2, and GFDL models), whereas other models exhibit convection only in the Irminger and GIN Seas (e.g., the CanESM5, ACCESS, and MIROC models). However, no model exhibits

convection exclusively in the Labrador Sea and not in the Irminger or GIN Seas.

Regarding the second point about the connection of this North Atlantic MLD spatial pattern to AMOC weakening in response to forcing, Figure R6 below shows the inter-model correlation between preindustrial MLD and AMOC change under 1pctCO2 forcing. Figure R6 reveals an anticorrelation between preindustrial MLD and forced AMOC change in the Labrador and Irminger Seas, i.e., since the sign of AMOC change is negative (a decrease in strength), models with deeper baseline MLD in these basins have a stronger AMOC decline under $CO_2$ forcing. This is consistent with our result in Fig. 3b that deeper baseline MLD in this region is associated with stronger baseline AMOC, together with the fact that a stronger baseline AMOC is associated with a stronger forced AMOC decline (e.g., Lin et al. 2023). There is also a small area of positive inter-model correlation in the GIN Seas (Fig. R6) where deeper mixed layer depths are associated with reduced AMOC decline, although the small extent of this area makes this relationship statistically less robust than the opposite signal found in the Labrador and Irminger Seas, and its physical interpretation is unclear.

[Figure]

Figure R6: Map of inter-model Pearson correlation coefficient across 28 CMIP6 models between local preindustrial annual mean mixed layer depth and AMOC change in years 60–80 of the 1pctCO2 experiment relative to preindustrial.

Since our study systematically uses AMOC strength only as a preindustrial baseline parameter and only focuses on the ocean heat uptake efficiency as a future forced response, we consider these results to be out of the scope of our study and elect not to include them in the main manuscript.
* * *
**Comment 1.3**

Line 42: add "in a historical context" to distinguish from future scenarios
* * *
**Response:**

Added as suggested.

**Comment 1.4**

Line 148: Specify what time period of the preindustrial control run do you include in the averaging? Is it only the period preceding the branching, the period starting af the same time as the branching, or the full preindustrial control simulation?

**Response:**

We have now added the following sentence:

"The model fields used as input to the EOF analysis are preindustrial upper-ocean stratification and mixed layer depth averaged over the time period in the preindustrial run corresponding to the first 150 years of the 1pctCO2 experiment used to determine OHUE in each model."

**Comment 1.5**

Lines 150-151: Does this process leave you with 23 models, or you get 28 models left after the elimination of outliers?

**Response:**

This outlier removal process leaves us with 23 models in the EOF analysis, we have now clarified this:

"For this, the EOF algorithm was iteratively applied five times to the preindustrial annual mean MLD fields of all models and the model with the most extreme value of the first principal component was removed, leaving a total of 23 models for the final EOF output. For all other non-EOF analyses in this study, the full set of 28 models is used. "

**Comment 1.6**

Line 158: 28 models before or after the removal of outliers?

**Response:**

We generally use the full set of 28 models for all analyses except the EOF analysis. Thus, the number of models given in this sentence is indeed correct. We have further clarified this in the paragraph cited in the response to comment 1.5 above.
* * *
**Comment 1.7**

Lines 518-520, this manuscript is now published, and the updated reference is: Gregory, J.M., Bloch-Johnson, J., Couldrey, M.P. et al. A new conceptual model of global ocean heat uptake. Clim Dyn 62, 1669–1713 (2024). https://doi.org/10.1007/s00382-023-06989-z
* * *
**Response:**

We have now updated this reference to its most recent version.

**References**

Heuzé, Céline and Hailong Liu (2024). "No Emergence of Deep Convection in the Arctic Ocean Across CMIP6 Models". en. In: *Geophysical Research Letters* 51.4, e2023GL106499. ISSN: 1944-8007. DOI: 10.1029/2023GL106499. URL: https://onlinelibrary.wiley.com/doi/abs/10.1029/2023GL106499 (visited on 03/07/2025).

Lin, Yuan-Jen, Brian E. J. Rose, and Yen-Ting Hwang (Feb. 2023). "Mean state AMOC affects AMOC weakening through subsurface warming in the Labrador Sea". EN. In: *Journal of Climate* -1.aop. Publisher: American Meteorological Society Section: Journal of Climate, pp. 1–44. ISSN: 0894-8755, 1520-0442. DOI: 10.1175/JCLI-D-22-0464.1. URL: https://journals.ametsoc.org/view/journals/clim/aop/JCLI-D-22-0464.1/JCLI-D-22-0464.1.xml (visited on 02/27/2023).

**2 Response to reviewer 2**
* * *
**General Comments**

Vogt at al. propose an original study reconciling past and recent literature focusing on the drivers of ocean heat uptake efficiency (OHUE), a major metric linked to the role of the ocean in mitigating climate change. Previously proposed drivers were either upper-ocean properties, meridional overturning metrics, or both, with limited understanding of the linkage between the two, particularly under Earth system modelling frameworks such as the Coupled Model Intercomparison Project (CMIP). Using an extensive ensemble of the last CMIP models, the authors identify the drivers of OHUE using global and regional approaches and explore the links between these drivers. Thus, the manuscript addresses particularly relevant scientific questions fitting the scope of the Ocean Science (OS) journal. The manuscript does not present any novel data or tools but definitely bring a very welcome and truly missing level of fundamental understanding by investigating the links between major ocean processes that are keys to improve our knowledge of the climate system. Substantial conclusions are reached, for example confirming previous findings on the role of upper-ocean stratification in the Southern Ocean on OHUE and clarifying the previously proposed role of the Atlantic meridional overturning circulation (AMOC).

Going through the review criteria of the OS journal:

The scientific methods and assumptions are valid and clearly outlined; the results are sufficient to support the interpretations and conclusions (see further down for suggestions); the description of experiments and calculations is almost perfectly complete, traceable, and precise to allow for reproducibility (see further down for suggestions); the authors give almost systematically proper credit to related work and clearly indicate their own original contribution (see further down for suggestions); the title clearly reflects the contents of the paper; the abstract provides an adequate, concise and complete summary; the overall presentation is well structured and clear, and so is the language that is fluent, particularly well-written, and precise, making it very pleasant to read, thank you very much for this. The mathematical formulae, symbols, abbreviations, and units are all correctly defined and used. The number and quality of references are also appropriate. No supplementary material is provided due to the use of appendixes. However, I still found some parts of the paper that should be clarified or improved, leading to relatively minor revisions.
* * *
**Response:**

We thank the reviewer for their positive evaluation and their thorough and helpful comments, which have substantially improved our manuscript. We have taken each comment into account, provided responses to each point below, and adapted the manuscript accordingly.
* * *
**Comment 2.1**

Use of different members from 3 models:

The Table A1 shows that different members were used for piControl and 1pctCO2 experiments in the 2 HadGEM3 models and the GISS model. Please correct if these are typos. If this is not typos, for these 3 models (10% of the model ensemble), the OHUE anomalies between 1pctCO2 and piControl experiments aiming at removing any model drift might generate artificial variability in OHUE in the analysed signal which could be of (negligible?) concern for statistic and signal processing methods. This is not the case for the application of EOFs limited to data from the piControl experiment, but these differences could alter the OHUE vs. driver correlations/relationships investigated. I presume that using different members between experiments for these models have been done to circumvent limitations in data availability of these models/experiments. If so, I suggest mentioning this aspect in the Methods section, ideally including an assessment of the implications.
* * *
**Response:**

We thank the reviewer for noticing this discrepancy.

In the case of the GISS-E2-1-G model, this was indeed a typographical error, and we have now corrected the 1pctCO2 member for this model to r1i1p1f1 in Table A1.

In the case of the two HadGEM3 models, the member labels given in Table A1 are correct, because the preindustrial "parent" experiment corresponding to these 1pctCO2 experiments (given by the "`parent_variant_label`" NetCDF attribute in the 1pctCO2 output files) is indeed r1i1p1f1. Therefore, we have left the entries in Table A1 for the two HadGEM3 models unchanged.
* * *
**Comment 2.2**

CO2-doubling OHUE vs. preindustrial ocean conditions:

The OHUE computed under doubling atmospheric CO2 conditions relative to preindustrial from the 1pctCO2 experiment is systematically compared to ocean properties and meridional overturning strengths under preindustrial atmospheric CO2 conditions. It would be interesting to duplicate this approach and determine if the relationships are similar using the same CO2-doubling period for ocean properties and meridional overturning strengths. My guess is that the relationships would be similar. If it happens that they are not similar, why?
* * *
**Response:**

We have now repeated the computations for Figure 1c–e and Figure 2 in the main manuscript using years 60–80 in the 1pctCO2 experiment instead of preindustrial means as the time period for stratification, MLD, AMOC, $M_{SO}$, and $M_{WMT}$; see Fig. R7 below. Note that the number of models used is slightly smaller due to data availability issues for the 1pctCO2 experiment.

Comparing the panels of Fig. R7 to Figs. 1 and 2 in the main manuscript, we can see that the spatial inter-model correlation patterns are very similar for stratification and MLD, although the signal in the subpolar North Atlantic is shifted southwards for stratification and slightly weaker for MLD. For the AMOC and $M_{SO}$, a significant positive correlation with OHUE persists even after 60–80 years of $CO_2$ forcing, but the correlation coefficient is lower than for preindustrial AMOC and $M_{SO}$ strength. For $M_{WMT}$, there is again no correlation with OHUE, as in the preindustrial case.

Overall, the relationships between the overturning cells (AMOC, $M_{SO}$, and $M_{WMT}$) and upper ocean metrics (stratification and MLD) on the one hand and OHUE on the other hand remain remarkably similar when employing years 60–80 of the 1pctCO2 experiment as a "baseline" instead of the piControl experiment. This suggests that the relatively large inter-model differences in these quantities that are present in the preindustrial state mostly persist even under transient $CO_2$ forcing, even as the absolute value of these quantities changes (e.g., stratification increases and AMOC strength decreases). It is possible that these differences collapse at longer timescales and under stronger forcing.

Since, as the reviewer notes, we systematically use preindustrial quantities as potential controlling factors on OHUE in the main manuscript, we suggest not to include this additional analysis in the main manuscript, but we have added the following to the main text:

"The relationships shown in Figs. 1 and 2 still hold when averaging all variables over years 60–80 in the 1pctCO2 experiment instead of over the preindustrial experiment, but they tend to be weaker (not shown). This bolsters our results and our approach of focusing on preindustrial controls on OHUE."

[Figure]

Figure R7: Scatter plots and inter-model correlation maps between stratification, MLD, AMOC, $M_{SO}$, $M_{WMT}$ on the one hand, and OHUE on the other hand. All quantities were calculated as means over years 60–80 in the 1pctCO2 experiment.
* * *
**Comment 2.3**

Model ensemble vs. observation for bias assessment:

It seems that CMIP6 "historical" experiments have been used for comparison with the ECCO state estimate in Figure 4, but the use of such experiment does not seem to be mentioned in the Methods section. Please mention it if so, including the list of members. Or did the authors sampled the 1pctCO2 experiments on a period corresponding to the mean atmospheric CO2 levels found during ECCO's spanning period (1992-2017)? The latter seems inadequate considering the availability of the CMIP6 historical experiments.
* * *
**Response:**

In Figure 4, we indeed use the CMIP6 historical experiment to compare model output to state estimate data during the same time period (1992–2017). We have now added the historical members used for each model to Table A1 and have changed the first sentence of the Methods section to the following:

"We use model output from a set of 28 climate models from 14 modeling centers run in three CMIP6 experiments: a baseline experiment with preindustrial forcings (piControl experiment), a historical scenario with realistic forcing from 1850–2014 (historical experiment), and a perturbed scenario forced by an idealized atmospheric $CO_2$ increase of 1% per year during 150 years (1pctCO2 experiment)."
* * *
**Comment 2.4**

Line 86: "four questions"
* * *
**Response:**

Fixed as suggested.
* * *
**Comment 2.5**

Line 91: "idealized atmospheric CO2 increase"
* * *
**Response:**

Added as suggested.

**Comment 2.6**

Line 93: "model outputs"

**Response:**

Fixed as suggested.

**Comment 2.7**

Line 94: "are regridded"

**Response:**

Fixed as suggested.

**Comment 2.8**

Line 117: Are all CMIP6 models providing sea water density as an output? If not, I presume that you computed it from temperature and salinity, maybe with a toolbox such as TEOS-10. I suggest mentioning this aspect in section 2.2.

**Response:**

We have added a sentence above this paragraph to specify that we have indeed used TEOS-10 to compute density from temperature and salinity:

"Potential density for use in the definitions of mixed layer depth and the Southern Ocean upper overturning cell below is computed from ocean potential temperature and salinity using the TEOS-10 software toolbox (McDougall and Barker, 2011)."

**Comment 2.9**

Line 178: Bourgeois et al. (2022) did not link OHUE to *global* mean upper ocean stratification, but only to *Southern Ocean mid-latitude* upper ocean stratification.

**Response:**

It is true that Bourgeois et al. (2022) used Southern Ocean mid-latitude stratification, and not global mean stratification, in their emergent constraint on OHUE. Our result in Fig. 1a showing no link between global mean stratification and OHUE does therefore not directly contradict the results of Bourgeois et al. However, our result could still "at first sight appear to contradict" the findings of Bourgeois et al. to a general reader, which is why we propose to keep the formulation of this sentence as it is. The intention here is not to discredit the findings of Bourgeois et al. (in fact they are perfectly coherent with our results), but rather to motivate a closer look at regional (instead of global mean) stratification.

**Comment 2.10**

Line 182: "(Fig. 1a)"

**Response:**

Fixed as suggested.

**Comment 2.11**

Line 184-186: I suggest mentioning the agreement with previous findings of the particularly strong link between OHUE and upper-ocean stratification in the Southern Ocean mid-latitude (Bourgeois et al., 2022; Liu et al. 2023).

**Response:**

We have now added the following sentence:

"This is consistent with previous studies showing a strong link between OHUE and upper-ocean stratification in the mid-latitude Southern Ocean (Bourgeois et al., 2022; Liu et al., 2023)."

**Comment 2.12**

Line 338-339: Suggest removing the parenthesis delimiting the last sentence.

**Response:**

Removed as suggested.

**Comment 2.13**

Line 439: Please include acknowledgements to WCRP/CMIP/ESGF, and any mandatory disclaimer from the EU grant agreement, if so.

**Response:**

We thank the reviewer for the reminder and have now added a standard WCRP/CMIP/ESGF acknowledgement.

**Comment 2.14**

Figure 1: Panels a-b shows 26 dots in accordance with the ensemble size. However, the figure's legend seems incomplete with only 21 models (of the last panel I presume). Please show the full legend.

**Response:**

This has now been corrected and the legend in Figure 1 now shows all 28 models.

**Comment 2.15**

Figures with maps: Please add longitude/latitude coordinates and consider using a common colorbar for several panels where appropriate: e.g. a single colorbar could be used in Figure 2, 3, 5, A1, A2, A4-6, A8, and A9. Please also consider increasing the font size of many of the very tiny panel's titles from many of the appendixes figures.

**Response:**

We have now added longitude/latitude labels and/or shared colorbars among panels to Figures 2, 3, 4, 5, A1, A2, and A6.

We have also increased the font size used in Figures A1, A2, and A6.

**Comment 2.16**

Figure 6: I suggest using the same acronyms as earlier in the manuscript: $M_{\text{SO}}$ and $M_{\text{WMT}}$ instead of $\text{MOC}_{\text{SO}}$ and $\text{MOC}_{\text{WMT}}$.

Thank you very much for addressing my comments.

**Response:**

Changed as suggested.
We thank the reviewer for their helpful comments.